# Multielectrode Arrays for Functional Phenotyping of Neurons from Induced Pluripotent Stem Cell Models of Neurodevelopmental Disorders

**DOI:** 10.3390/biology11020316

**Published:** 2022-02-16

**Authors:** Fraser P. McCready, Sara Gordillo-Sampedro, Kartik Pradeepan, Julio Martinez-Trujillo, James Ellis

**Affiliations:** 1Department of Molecular Genetics, University of Toronto, Toronto, ON M5S 1A8, Canada; fraser.mccready@mail.utoronto.ca (F.P.M.); sara.gordillosampedro@mail.utoronto.ca (S.G.-S.); 2Developmental & Stem Cell Biology Program, The Hospital for Sick Children, Toronto, ON M5G 0A4, Canada; 3Department of Physiology and Pharmacology, Department of Psychiatry, Robarts Research and Brain and Mind Institutes, Schulich School of Medicine and Dentistry, Western University, London, ON N6A 5B7, Canada; kpradeep@uwo.ca (K.P.); julio.martinez@robarts.ca (J.M.-T.)

**Keywords:** neurodevelopmental disorders, multielectrode arrays, induced pluripotent stem cells, neurons, astrocytes

## Abstract

**Simple Summary:**

Multielectrode array technology allows researchers to record the spontaneous firing activity of cultured neurons over a period of multiple weeks or months. These data can be valuable for understanding how the functional relationships between neurons evolve as they begin to form connections and wire into a functional network. This technology has been adopted by researchers using stem cells to produce human neurons in culture to study neurodevelopmental disorders. However, the dizzying complexity and scale of the data generated have posed some challenges with the analysis and interpretation of experimental results. Here, we first provide historical context as to why multielectrode array platforms were originally developed, and use this perspective to explore some of the challenges currently facing the field. We then highlight new analysis methods, provide some guidance for improving the analysis of multielectrode array data, and discuss standardizing how these findings are communicated in scientific publications.

**Abstract:**

In vitro multielectrode array (MEA) systems are increasingly used as higher-throughput platforms for functional phenotyping studies of neurons in induced pluripotent stem cell (iPSC) disease models. While MEA systems generate large amounts of spatiotemporal activity data from networks of iPSC-derived neurons, the downstream analysis and interpretation of such high-dimensional data often pose a significant challenge to researchers. In this review, we examine how MEA technology is currently deployed in iPSC modeling studies of neurodevelopmental disorders. We first highlight the strengths of in vitro MEA technology by reviewing the history of its development and the original scientific questions MEAs were intended to answer. Methods of generating patient iPSC-derived neurons and astrocytes for MEA co-cultures are summarized. We then discuss challenges associated with MEA data analysis in a disease modeling context, and present novel computational methods used to better interpret network phenotyping data. We end by suggesting best practices for presenting MEA data in research publications, and propose that the creation of a public MEA data repository to enable collaborative data sharing would be of great benefit to the iPSC disease modeling community.

## 1. Introduction

Induced pluripotent stem cell (iPSC) technology has afforded many benefits for the modeling and study of human neurodevelopmental disorders. In this system, somatic cell types such as skin fibroblasts, peripheral blood cells, and urinary epithelial cells can be collected from patients who already have a clinical diagnosis of the disorder under investigation and then reprogrammed into iPSCs through the delivery of pluripotency-associated transcription factors. Using well-established induction protocols, reprogrammed iPSCs can then be differentiated into functional neuronal and glial cell types of interest carrying the same genetic background as the patient they were originally derived from, preserving the complex genetic architecture associated with many neurodevelopmental disorders. They can also be used as a platform to evaluate the effect of genetic rescue intervention. In this way, researchers are afforded a renewable source of patient-derived neurons and glial cells for study and experimentation, without requiring any invasive biopsy of human CNS tissue. These in vitro cultures of iPSC-derived neuronal cells can then be utilized in molecular, morphological, and functional phenotyping experiments, as well as for investigating the effect of rescue genetic manipulations and preclinical drugs.

The past decade has seen increasing use of multielectrode arrays (sometimes referred to as microelectrode arrays or MEAs) for functional phenotyping experiments in iPSC disease modeling studies. In MEA systems, special tissue culture plates which contain a grid of small electrodes embedded in the bottom of each well are used to record extracellular action potentials from neuronal cells cultured directly on top of the recording electrodes. Simultaneous recording of spiking activity from multiple spatial locations within a culture allows for the interrogation of population-wide firing activity, and understanding of how network dynamics may be altered in disease states. Newer multi-well MEA systems, which come in 12, 24, 48, and 96-well formats, have been particularly favoured in disease modeling studies as they afford higher throughput than traditional single-cell recording techniques for functional phenotyping and drug screening applications. Moreover, as extracellular MEA recording is non-destructive to neuronal integrity, the same neuronal culture can be monitored over multiple weeks or months of in vitro development, potentially providing insights into neurodevelopmental processes which function at longer timescales.

The adoption of MEA technology by the iPSC disease modeling field is fairly new, and our intent with this review is not to provide a comprehensive overview of all reported results and disease phenotype from every recent MEA phenotyping study. Instead, we aim to draw attention to some of the challenges that come with deploying MEA technology in an iPSC disease modeling context, and highlight opportunities for improving their use. We begin by providing an overview of the historical background through which MEA technology has developed. We then discuss some of the more recent developments applying MEA technology to iPSC disease modeling studies, as well as the challenges that come with the analysis and interpretation of complex multichannel data.

## 2. Historical Perspectives

While MEA technology only began to see use in the field of iPSC disease modeling within the last decade, the development of the first MEA systems actually date back to the early 1970s [1,2,3]. We feel that a brief overview of the nearly 50 year history of this technology and the scientific questions that lead to its initial development will add a valuable layer of historical context that better highlights its strengths and potential applications in disease modeling research.

### 2.1. Developing Long-Term Culture Methods for Nervous Tissue

The development of in vitro MEA systems came in part as a response to significant advances in neural tissue culture techniques made throughout the 1950s and 1960s, as developmental neurobiologists sought to establish neural explant cultures as a simplified model system for studying early CNS development and tissue organization. Much of the initial work in this area was undertaken with the simple goal of investigating whether neurons were even capable of retaining their basic neurophysiological properties, such as the ability to generate action potentials, when kept isolated in long-term cultures. Prior attempts to culture neural tissues in the early decades of the 20th century had limited success in maintaining these cultures for longer than one or two weeks, and cells that did manage to survive past this time tended to develop abnormal morphologies uncharacteristic of neuronal cell types [4]. Consequently, there was a considerable, well-established concern that cultures of explanted neural tissue would not only fail to undergo biologically relevant, organotypic development in vitro, but were moreover doomed to a fate of “unavoidable de-differentiation” [4] (p. 84) and atrophy [5]. The ability of cultured CNS explants to organize functional synaptic networks capable of producing complex patterns of bioelectric activity was similarly approached with appreciable skepticism. Indeed, when an initial study published by Hild and Tasaki in 1962 failed to find evidence for synaptic interactions in cultures of rat cerebellar explants, the authors concluded that “*Neurons in tissue culture differ from those in vivo in one important aspect. A neuron in vivo is always part of a neural network, whereas a neuron in tissue culture no longer has synaptic connections with other neurons*” [6] (p. 300).

It will perhaps be unsurprising for present-day neuroscientists to learn that these concerns of neural de-differentiation and synaptic isolation in long-term cultures proved to be unfounded. Numerous studies by Drs. Stanley Crain, Murray Bornstein, Edith Peterson, and others demonstrated that CNS explant cultures retained their electrophysiological functions for months in vitro, often developing complex patterns of synaptically-mediated bioelectric activity including rhythmic, oscillatory discharges reminiscent of in situ recordings (reviewed in [7]). In one particularly surprising study by Crain and Bornstein (1972), these complex oscillatory activity patterns were recorded from “reaggregated” clusters of CNS neurons that had been randomly dispersed into the culture chamber as a suspension of single cells, rather than an explant of highly organized tissue [8]. Moreover, the spontaneous activity patterns that developed in these cultures often appeared to be synchronized, even when recording from reaggregates separated by a distance of up to 3 mm. These findings suggested that such cultures of dissociated CNS neurons have an intrinsic capacity to self-organize into functional networks capable of generating complex patterns of organotypic activity independent of the complex anatomical and histological organization found in many CNS tissues. The authors touch on the implications of these findings for future research in the concluding sentence of their paper; that careful and continuous study of the combined morphological and electrophysiological characteristics of many dissociated neurons throughout the process of re-establishing de novo connections might allow researchers to untangle the unique roles that individual cells play in generating intricate patterns of coordinated network activity.

### 2.2. Early MEA Technology

This idea of continuously monitoring many different cells within a culture to understand how the myriad electrical interactions between individual neurons self-organize into complex synaptic networks was a key motivation behind the development of in vitro MEA platforms. In another 1972 paper, Thomas et al. expressed similar thoughts to those of Crain and Bornstein, suggesting that the most interesting scientific questions to explore with newly established long-term tissue culture models were those which asked how the unique bioelectrical relationships between individual cells might develop and evolve with time [1]. The development of the first MEA platform was a directed effort to overcome some of the technical barriers in the way of answering these questions, by attempting to provide researchers with “*a convenient, non-destructive method for maintaining electrical contact with an individual culture, at a large number of points, over a period of days or weeks*” [1] (p. 61). To accomplish this, the fabrication of this MEA system utilized photolithography techniques developed for circuit-board etching in the microelectronics industry to “print” a 2 × 15 array of electrodes onto a glass coverslip, which would then serve as the bottom of a culture chamber. Dissociated cells or tissue could then be cultured directly overtop of the printed microelectrodes. Importantly, this technique of “bringing the cells to the electrodes” had a considerable advantage when it comes to continuous monitoring of in vitro activity over long time periods, as conventional in vitro recording techniques of the era often required experimenters to expose cultures to open, non-sterile conditions while recording electrodes were inserted in the culture chamber. After being exposed to these open recording environments, it was difficult to keep cultures alive for more than a day, meaning these recordings were often endpoint experiments [9].

Initial experiments with Thomas et al.’s device in 1972 attempted to record the electrical discharges from dissociated embryonic chick heart tissue, and were ultimately successful in detecting rhythmic patterns of ECG-like electrical potentials which co-occurred with the rhythmic mechanical contractions of cardiac cells observed in the culture [1]. The first in vitro MEA recordings of neural tissue would come two years later by Shtark et al. (1974), who successfully recorded EEG-like field potentials from embryonic and early postnatal rat brain explants using a similar photoetched MEA device, but were unsuccessful in recording single-unit spikes [2]. Improvements to planar MEA design and fabrication made by Dr. Guenter Gross et al. in 1977 allowed for the detection of single-unit spikes for the first time, recording from parietal ganglia explants of *Helix pomatia* [3]. Additional follow-up studies by Gross et al. further demonstrated the ability of this device to record single-unit activity from three different electrodes simultaneously, as well as the ability to record the same culture of dissociated neurons for a period of multiple days [10,11,12]. While simultaneous recording from all 36 available electrodes was technically possible, the large number of amplifiers that would be required to handle signal processing (one for each channel) along with the daunting challenges posed by the analysis and interpretation of this quantity of information would have made this tremendously impractical in practice. Indeed, the authors are careful to warn that “*The formidable amount of data which can be gathered by 36 active electrodes will challenge the current capabilities of data analysis*” [11] (p. 67), and “*…the difficulties of handling the data that can be gathered by 36 electrodes should not be underestimated*” [11] (p. 67)—sentiments that will likely still resonate with many current MEA users nearly 40 years later.

### 2.3. Qualitative Descriptions of Network Activity

Continued work with in vitro MEA systems throughout the 1980s and early 1990s was limited to a small number of research groups. This was primarily due to the lack of commercially available MEA systems, as well as the large amounts of equipment and computing power required for multichannel data handling, storage, and analysis. Much of the work completed during this time was devoted to simply describing the spontaneous activity patterns observed in MEA cultures, and developing meaningful *qualitative* descriptions of the different characteristic firing motifs that often appear in cultured networks (see [13,14,15] for thorough review). While modern researchers now have abundant access to cheap computing power and more sophisticated spike train analysis methods for quantifying aspects of network behaviour, we believe that reintroducing these qualitative descriptors to the iPSC disease modeling community will nonetheless be of some benefit to the field. In particular, we hope this terminology will help provide a shared vocabulary for researchers to use in describing the broad phenotypic differences in activity patterns displayed by different control and disease-state networks, the complexity of which is often difficult to convey in simple terms.

Based on their extensive early experience with MEA recordings from cultures of dissociated murine embryonic spinal cord cells, Guenter Gross and colleagues suggested that all patterns of network activity can be loosely classified into one of six different characteristic “modes” of firing activity based on simple visual inspection of recorded spike patterns. These modes, numbered in order of increasing spiking frequency, are: M1—no spiking, M2—random low-frequency spiking, M3—patterned spiking with weak bursting, M4—patterned bursting, M5—periodic bursting, and M6—burst fusion leading to continual high-frequency spiking [14]. Observing mouse spinal cord cultures over a sufficiently long recording period (multiple hours to days), it was seen that cultured networks would gradually transition between multiple different activity modes during the course of the recording, ranging from no spiking (M1) through to periodic bursting (M5). Gross would define this “relative network dynamic range” as “*the range of activity states or modes a network can attain spontaneously or during any one experimental treatment*” [15] (p. 306). Addition of ethanol or GABA to the culture media restricted the relative network dynamic range to lower activity modes (M1–M3), while the addition of strychnine and NMDA restricted the relative network dynamic range to higher activity modes (M4–M6) [15].

It is worthwhile noting that the typical recording periods used in current iPSC disease modeling studies is much shorter than those described above, most commonly ranging between 5 and 10 min. If networks of iPSC-derived neurons also have a natural tendency to spontaneously transition between different modes of network activity over the course of multiple hours, these short recording intervals likely only capture a sampling of the different activity modes contained in that network’s relative dynamic range. Anecdotally, our lab has observed that during time course experiments, networks which seem to have settled into stable patterns of rhythmic bursting activity will, on rare and sporadic occasions, appear to revert back to less structured modes of firing activity at seemingly random timepoints in the series. These “reversion events” often appear as outliers and can add a considerable amount of noise to time course datasets. Moreover, the presence or absence of reversion events could also be a meaningful phenotypic signal reflecting higher-order changes in network dynamics. Whether recording networks of iPSC-derived neurons for longer durations will reveal long-term temporal variations in network activity remains to be reported.

### 2.4. Quantitative Descriptions of Network Activity

Early attempts to comprehensively characterize network activity patterns using a large number of different quantitative metrics can also be tied back to the work of Guenter Gross and colleagues. Seeing that the activity patterns of cultured networks were highly sensitive to minute changes to the chemical composition of culture media, Gross et al. (1992) saw the potential to utilize in vitro neuronal networks coupled to MEA devices as biosensors for detecting and identifying substances such as drug and toxins [16]. Much like current disease phenotyping studies which aim to extract a large number of qualitative spike train metrics from MEA recordings and identify those which might serve as biomarkers for a certain neurodevelopmental condition, the goal of early work in this area was to define quantitative metrics of network activity dynamics and select those which might serve as markers for identifying and classifying different substances.

In work that shares a surprising resemblance with many present-day network phenotyping assays, Gramowski et al. (2004) aimed to characterize the unique “substance-specific response profiles” of in vitro networks exposed to six different pharmacological conditions [17]. To do this, the authors derived 30 different statistical features which describe different aspects of the network activity. These included several commonplace activity metrics such as burst frequency, burst duration, spike rate, and interspike interval (ISI) measures, as well as some unique coefficient of variation metrics, CV_NETWORK_ and CV_TIME_, which together to capture a coarse-grained picture of how network activity is organized both spatially and temporally. Specifically, CV_NETWORK_ quantifies the degree to which firing activity is synchronized across all electrodes in the network, while CV_TIME_ quantifies the periodicity of firing activity across each recording channel. Both CV_NETWORK_ and CV_TIME_ were found to be invaluable quantitative metrics for discriminating between networks with different spatiotemporal behaviour, as they make effective use of the wealth of information contained within the organizational structure of multichannel neural recordings.

Many of these MEA analysis methods originally developed for use in screening of pharmacological compounds and toxins are highly transferrable to disease modeling applications, and interested readers may find cross-pollination between these related fields to be fruitful. More comprehensive reviews on the use of MEA systems in neurotoxicology applications can be found in [18,19].

## 3. Generating Neurons and Glia for MEA Phenotyping Assays

The discovery of mature cell reprogramming in the mid 2000s allowed the isolation of induced pluripotent stem cell (iPSC) lines from patients. Further progress in this field allowed differentiating human iPSCs into tissue-specific lines (e.g., neurons) for cellular phenotyping. For the purpose of this review, we will focus on production of 2D human cortical neurons culture to model circuitry dysfunction in the neurodevelopmental disorders. For recent reviews which discuss the use of MEA technology with 3D organoid cultures, see Pelkonen et al. (2022) and Passaro and Stice (2021) [20,21]. An overview of MEA studies using iPSC derived 2D human neurons is found in Table 1.

### 3.1. Neuronal Differentiation Methods for Generating Functional Neurons

While a multitude of distinct differentiation protocols exist for generating functional neuronal cell types from human iPSC lines, the majority of these can be broadly categorized as either directed differentiation protocols, or transcription factor programming methods. In commonplace directed neuronal differentiation protocols, iPSCs are first lifted from 2D monolayer cultures and allowed to aggregate into three-dimensional structures known as embryoid bodies (EBs). Various mitogenic and morphogenic compounds are then carefully added to cell culture media over a period of multiple weeks, mimicking extracellular signalling cues involved in normal embryonic development [44,45]. In particular, compounds such as SB431542 and dorsomorphin are commonly used to inhibit TGFβ and BMG signalling pathways, driving iPSCs to adopt a neural precursor cell (NPC) fate. Wnt signalling inhibitors such as XAV939 are also used to promote differentiation into more dorsal forebrain cell types, which are of interest in many neurodevelopmental disorders [44,45]. Once NPC cultures are established, differentiation into cortical neurons is achieved by supplementation of cell media with neurotrophic factors such as BDNF and GDNF, and the removal of FGF2 [46]. The end result of directed differentiation though such “dual-SMAD inhibition” protocols is a heterogenous culture of post-mitotic neurons as well as glial cells. Indeed, directed differentiation protocols targeting dorsal forebrain cell types produce mixed neuronal cultures which express cell markers for all six cortical layers, as well as markers for both excitatory and inhibitory neurons [47].

One significant downside of directed differentiation for producing MEA cultures is the relatively slow speed of neuronal maturation. It takes approximately 4 weeks to establish NPC cultures from iPSCs, and an additional 4–12 weeks of differentiation for neurons to mature into electrophysiologically active cells [46,47,48]. In our experience, cultures of iPSC-derived neurons can often be difficult to maintain in culture for longer than 9 or 10 weeks from the NPC stage, as issues with poor cell adhesion can cause entire neural monolayers to detach from the substrate. MEA cultures seem to be particularly vulnerable to this issue, and media changes must be undertaken with great care, especially at later timepoints.

More recently developed methods of neuronal differentiation drive cell fate conversion by utilizing an inducible transgene expression system to force the overexpression of pro-neurogenic transcription factors such as Neurogenin 2 (Ngn2) in iPSCs [49]. In comparison to directed differentiation approaches, transcription factor programming-based differentiation protocols allow for much more rapid production of neurons for functional phenotyping experiments by bypassing the transition through an NPC stage, converting iPSCs to neurons in a single step. Neuronal differentiation via forced expression of *Ngn2* produces a more homogenous population of excitatory layer 2/3 cortical neurons which begin to show signs of electrophysiological activity in as little as one week after inducing transgene expression [49]. Neurons generated through this method generally require co-culture with an additional source of astrocytes in order to promote proper neuronal maturation. Primary cultures of either mouse, rat, or human astrocytes are commonly used for MEA co-cultures; however, improvements in astrocyte differentiation protocols are opening the door to the possibility of using genotype-matched iPSC-derived astrocytes in future MEA experiments (discussed below).

In addition to excitatory neurons, additional transcription factor programming methods have been developed to produce homogenous populations of inhibitory neurons, relying on the forced expression of *Ascl1* and *Dlx2* instead of *Ngn2* [50]. This ability to rapidly produce both excitatory and inhibitory cell types provides neurodevelopmental disease modeling researchers a unique opportunity to study how different ratios of excitatory to inhibitory cell types in MEA cultures might impact the development of network activity. During development, GABAergic synapses are established by an increase in the KCC2:NKCC1 ratio of chloride co-transporters, shifting the chloride reversal potential from depolarizing (excitatory) to hyperpolarizing (inhibitory). Disruptions in the excitatory (E) and inhibitory (I) ratio haven been suggested to occur in Rett syndrome (reviewed in Ip et al. 2018), and have been studied extensively in the context of autism spectrum disorders (ASD) and schizophrenia (SCZ) [51,52,53,54]. Recent work by the Nadif-Kasri lab used MEAs to characterize a network-level phenotype for Kleefstra syndrome (KS), a neurodevelopmental disorder caused by mutations in the methyltransferase EHMT1, and found indications of E/I imbalances [39]. By generating an excitatory only model using isogenic iPSC-derived neurons from KS patients, they showed that KS neurons developed longer network bursts driven by the upregulation of the NMDAR subunit NR1. Subsequently, they rescued NMDAR hyperexcitability by pharmacological inhibition of NMDAR activity, opening up new therapeutic avenues for KS. Recently, the same group described a new fast differentiation protocol to generate GABAergic neurons by overexpressing Ascl1 and forskolin (FSK) [55]. Using this protocol, they established functional networks containing excitatory (Ngn2) and inhibitory (GABAi) neurons in MEAs able to replicate the GABAergic hyperpolarizing shift mimicking network maturation. Novel protocols such as this one could be applied to patient-derived iPSCs to generate mature, heterogeneous and functional networks, providing a new and more complex platform to study network formation and electrophysiological imbalances, keeping a human genetic background.

### 3.2. iPSC-Derived Astrocytes and Genotype-Matched Co-Cultures

In general, to establish functional neuronal networks on MEA, neurons need to be plated together with supporting glial cells. Commonly, mouse astrocytes are used to enable network formation and maturation, and to be able to keep these plates for long culturing periods [56,57]. In recent decades, interest has shifted to generating astrocytes from human iPSCs after finding evidence of glial cells playing more important roles than just being the supporting cells for neurons [58,59]. Early in vitro studies from the Barres group showed that adding exogenous astrocytes to neuronal primary cultures accelerated their development, shortening the culturing time needed for spontaneous network formation prior to activity recording [56]. This research demonstrated that astrocyte–neuron communication is paramount for network formation, maintenance and maturation, and provided novel strategies for in vitro culturing. Zhang et al. 2016 showed that human astrocytes express a subset of specific genes not found in mouse, which indicates and supports functional differences between species [60]. In addition, recent single-cell (sc) RNAseq experiments have shown that human iPSCs can differentiate into heterogeneous astrocyte populations as seen in vivo with distinct subtypes that can be classified according to maturation status and function [61]. Moreover, single-cell in situ hybridization experiments in the mouse cortex and hippocampus showed that these functional subtypes could be mapped to specific regions [62,63], further highlighting the importance of using same-species and same genotype cells when studying network phenotypes. The rare availability of postmortem brains from individuals with neurodevelopmental disorders such as Rett syndrome and ASD showcases the potential of isogenic iPSC lines as a renewable source of human neuron and astrocyte networks from the same patient, making them better matched comparisons.

Patch clamp and, more recently, MEA systems have been used to evaluate astrocyte contribution in models of ASD and Rett syndrome, showing astrocytes have non-cell-autonomous effects on neural networks driven by physical interactions and extracellularly released factors, which might not be captured in hybrid systems [22,64,65]. However, generating astrocytes in vitro has proven more complicated than making neurons because astrocyte genes are epigenetically suppressed until neurogenesis is underway [59,66,67]. Left to their own devices, progenitor cells eventually differentiate to astrocytes spontaneously after weeks in culture as it happens in vivo, generating mixed cultures that can be plated as a monolayer or kept as 3D structures with the option to isolate specific cell types with immunopanning techniques [61,64,65,66,67,68,69]. Despite generating cortex-like structures with cell type heterogeneity, these approaches rely on long culture periods to obtain high percentages of astrocytes, and isolation steps that require the expression of markers such as Hepa-CAM which appears after about 100 days in culture [68]. Alternatively, to pattern differentiation towards astrocytes only, the general approach is to activate and modulate the JAK-STAT pathway, which is responsible for gliogenesis during development [59,70]. By adding growth factors to the culture media such as CNTF, LIF and BMPs, progenitor cells can undergo terminal differentiation generating functional astrocytes [71,72,73,74,75,76,77,78,79,80] which can be regionally specified in the progenitor stage [72,79], slightly shortening culture time. With the aim to speed up these protocols so that establishing co-cultures and functional networks becomes easier and more feasible, other groups have focused on developing inducible transgenes to transiently overexpress combinations of JAK-STAT transcription factors NFIB, NFIA or SOX9 in fibroblasts to generate astrocytes in just over a week with varying efficiencies [81,82,83]. Despite being faster, these protocols require induction steps that often need optimization, and the selection of primed progenitor populations using FACS before terminal differentiation begins [83]. Moreover, to improve cell viability and yield, some protocols rely on the use of fetal bovine serum (FBS) [68,77,79,80,84] shown to affect astrocyte phenotypes by inducing a reactive status, thus potentially interfering with the observation of specific disease features [60,84].

Fast methods to generate astrocytes from NPCs have shown promising preliminary results when co-cultured with Ngn2 neurons in MEA plates, establishing functional neural networks, and opening the door to the use of same genotype cells within one culture. Russo et al. 2018 [22] established MEA co-cultures of human iPSC-derived neurons and astrocytes isolated from neurospheres from age-matched healthy controls and ASD patients. Despite not using isogenic lines, they found reduced spontaneous spike rates in ASD networks cultured in MEAs, and showed that co-culturing ASD isolated neurons with WT astrocytes improved some morphological and synaptic features of ASD neurons. In addition, Taga et al. 2019 established for the first time a co-culture of isogenic iPSC-derived neurons and astrocytes patterned to spinal cord identity and showed that astrocytic presence in the MEA as well as their density and the time of plating influenced neuronal morphology, electrophysiology and molecular properties [64]. These lines of research showcase the potential and suitability of using human only networks for disease modelling. Because the methods to functionally characterize astrocytes’ ability to support neurons in network setups have not been standardized, understanding and reporting in detail similar MEA parameters taken in the same conditions bring the opportunity for the field to generate reproducible data. See “Conclusions and Recommendations” for further discussion of standardizing MEA parameters and their presentation.

## 4. Challenges with Current Approaches to MEA Phenotyping

Within the field of iPSC disease modeling, MEA technology has primarily been deployed as a platform for performing functional phenotyping assays, with the goal of experiments being to define a network-level phenotype for the disorder under investigation by comparing the activity of affected neurons with that of healthy controls and looking for biologically relevant differences in network function. Once established, a well-defined phenotype can inform researchers about potential underlying molecular or cellular disease mechanisms, or can be utilized as a point of reference for subsequent phenotypic rescue experiments using genetic or pharmaceutical interventions. Modern multi-well MEA systems are particularly well suited for this purpose, as higher-throughput 48- and 96-well plate formats capable of recording many individual networks in parallel can be leveraged as a platform for rapid functional phenotyping and small candidate-based drug screens. In addition, commercial distribution of user-friendly software packages has streamlined the data analysis process by automatically extracting upwards of 50 different metrics which quantify many different aspects of network firing behaviour. This allows researchers to adopt a “shotgun approach” to functional phenotyping, where comparisons between control and disease conditions can be made for multiple different outputs in parallel. Despite the significant improvements in rapid data collection and analysis afforded by modern MEA systems, there are still some important challenges to address when it comes to the analysis, interpretation, and presentation of MEA phenotyping data.

### Appropriate Selection of Phenotyping Metrics

The success of any phenotyping assay in furthering our basic understanding of underlying disease mechanisms, or in generating meaningful endpoints for drug discovery, is ultimately reliant upon the appropriate choice of assay readouts by the experimenter. Ideal cellular phenotyping readouts should have clear, unambiguous interpretations which provide a logical direction for future mechanistic studies by highlighting cellular functions or processes that are likely disrupted in the disease state. In addition, established best practices for the design of phenotypic assays in translational research such as drug discovery suggest that optimal assay readouts should have clear biological relevance and be related to clinical endpoints of the disorder under investigation [85].

With this is mind, the high degree of complexity and spatiotemporal organization found in the patterns of network activity generated by in vitro networks is a significant challenge when it comes to selecting meaningful phenotyping metrics. This dynamical complexity means that phenotypic differences in the distinct activity patterns observed between disease and control networks often require a large number of different quantitative metrics to capture adequately, with each individual metric likely capturing a small piece of the disease phenotype. For example, even networks which have identical rates of firing can display significant variation in how those spikes are ordered temporally and distributed spatially. To illustrate this, we generated example raster plots of activity from 12 electrodes in four MEA wells which each contain exactly 1200 spikes in the span of 60 s for identical mean firing rates (MFR) of 20 Hz (Figure 1). However, through simple visual inspection of the raster plots, we can see that quantification of firing rate alone does not adequately capture differences in the activity generated within these networks, as they vary substantially in how that activity is ordered. The examples range from a single active electrode in a long burst (Figure 1A), to stochastic spiking at all electrodes (Figure 1B), multiple electrodes producing independent bursts (Figure 1C), and finally synchronized network bursts across multiple spatial locations (Figure 1D). Importantly, these variations in the temporal organization of network activity can be driven by variations in underlying neurophysiological processes [86,87,88,89,90]. As such, defining a network phenotype by any single network measure in isolation is likely to omit crucial information about how the spatial-temporal organization of spiking activity is disrupted in disease networks, which could provide important clues for identifying underlying neurophysiological functions driving pathologic activity patterns.

Multiparametric approaches to defining disease phenotypes can present their own issues when it comes to drawing meaningful interpretations about underlying disease mechanisms. While defining a disease phenotype in terms of many different activity metrics captures more information about how network dynamics are altered in disease states and is preferable to single-metric phenotyping, the subsequent task of drawing interpretations about how many disparate network metrics might be driven by a common disease mechanism can become so complex as to be nearly intractable. Multiparametric disease phenotypes also can create issues when it comes to phenotypic rescue experiments and drug discovery. Phenotypic endpoints that are defined by multiple different measures can create situations where a candidate drug compound may rescue some aspects of disease-associated patterns of network activity, while leaving others unchanged. Without a clear understanding of which metrics are most physiologically relevant to the disorder under investigation, this can create ambiguity when trying to evaluate and compare the efficacies of different interventions in preclinical drug screening assays. For instance, it might be unclear whether a drug treatment which rescues aberrant network burst frequency but not burst duration is more likely to lead to positive clinical outcomes than a drug which rescues burst duration but not burst frequency. This is an especially salient issue when it comes to modeling neurodevelopmental disorders such as ASD which are diagnosed solely on the basis of behavioural evaluations, as a large explanatory gap exists between the complex patterns of spiking activity that develop in in vitro network models and the complex patterns of human behaviour that characterize the disorder.

It is worthwhile noting that similar points to these have already been raised in the primary neuron literature, again by groups interested in utilizing in vitro MEAs as a platform for screening and identifying neurotoxic compounds. Indeed, a 2010 review paper by Johnstone et al. notes that while many previous studies in this area have focused on changes in average spike rate, this may not be the most descriptive metric that can be extracted from recording the data [19]. Instead, the authors emphasize the benefits of multiparametric data analysis methods and the importance of considering features such as synchronicity, burst structure, and burst rate, which capture more information about the spatiotemporal structure of underlying activity patterns. The use of feature selection methods for identifying which of the many extracted activity features are most informative for classifying a given substance is also discussed.

## 5. Expanding the MEA Analysis Toolkit in iPSC Disease Modeling

### 5.1. Computational Modeling Approaches and Analysis Methods

More recent attempts to address these challenges in the iPSC disease modeling field have focused on using more intricate computational analyses and in silico network simulations to reduce the number of measures required to capture relevant changes in network activity, or provide predictions about relevant neurophysiological processes that could be driving aberrant network behaviour. Borrowing from the field of dynamical system theory, Amataya et al. applied minimum embedding dimension (MED) analysis to phenotype complex activity patterns from control and ASD networks [27]. This analysis treats in vitro networks as complex dynamical systems whose different output spiking patterns or “states” can be described using a finite number of differential equations. The MED algorithm estimates the number of equations that are required to generate these activity patterns which is a measure of dynamical complexity, with more complex activity having a greater MED value. Initial analysis of MEA recordings found no difference in MFR between ASD and control networks. Instead, the patterns of activity generated by ASD networks were primarily characterized by variations in the organization of spike timing rather than overall spiking rate. Accordingly, these differences in firing pattern complexity were captured by the MED analysis, with the ASD networks having significantly lower MED values than controls. Intriguingly, the authors also found a significant correlation between the MED values and relevant clinical endpoints for ASD; in vitro neuronal networks with low MED values were more likely to be derived from individuals with lower non-verbal IQ and Vineland adaptive behaviour scores. It is interesting to note that reduced signal complexity has also been found in EEG recordings from individuals with ASD, and it has been suggested that the loss of complexity from physiological signals is hallmark shared by many clinical disorders [91,92,93,94].

Additional studies have employed the use of computational network simulations to aid in the interpretation of MEA phenotypes. Trujillo et al. 2021 utilized a computational model to predict whether rescue of reduced synapse numbers commonly observed in iPSC models of Rett syndrome would, in isolation, be sufficient to rescue alterations in functional network activity [95]. After confirming that increasing synapse numbers to wild-type levels is sufficient to rescue alterations in network firing rate in silico, the authors conducted an imaging-based screen of 16 different compounds expected to increase synapse numbers in vitro. Drug compounds which showed success in elevating synapse numbers in this imaging assay were then trialed in follow-up MEA experiments, which confirmed that both identified compounds were able to rescue altered network activity as well. Mok et al. (2021) utilize a similar in silico model to aid in the characterization and interpretation of Rett syndrome MEA phenotyping data [96]. Biophysical simulations showed that networks of adaptive leaky integrate-and-fire neurons constructed with Rett-like intrinsic membrane properties generate patterns of network activity which mirrored the phenotypes seen in real in vitro MEA recordings. Substituting wild-type Na^+^ and K^+^ channel current values into the Rett network model was sufficient to ameliorate the Rett-like network activity in simulations, suggesting that altered function of these channels could be a key mechanism driving the altered network-level behaviour. Together, these studies highlight how computational modeling approaches can be powerful tools for generating tractable hypotheses about underlying disease mechanisms from complex, multiparametric MEA data.

Another potential opportunity to improve MEA phenotyping studies of neurodevelopmental disorders lies in placing a greater emphasis on investigating differences in the trajectories of network dynamics as they develop between different recording periods. Much of the current literature reports phenotyping metrics at a single timepoint, despite having recorded network activity over the course of multiple weeks. Moreover, studies which do report data from multiple timepoints rarely focus on differences in trajectories as being potentially informative phenotypic differences. Such differences in the ways that network activity patterns might develop over a recording time course might reflect differences in neurophysiological processes which function at timescales longer than an average recording period, such as synaptogenesis and homeostatic plasticity mechanisms [97]. It may also be important to consider how variables measured at different timepoints contribute to network dynamics and how they interact with one another.

Emergent techniques using dimensionality reduction and state space analyses are commonly used in the analysis of multielectrode recordings taken in behaving animals, and have been applied to recordings of spontaneous activity in brain organoids [21]. Advances in machine learning may also aid with the analyses of multidimensional datasets emerging from MEA [98]. Poli and coworkers have summarized a series of approaches that can be used to analyze network connectivity using MEA recordings and suggested a holistic approach can be useful in identifying unique topological properties of neuronal networks that emerge from the interactions amongst different nodes [99]. Spencer and coworkers have used modeling approaches that use the multiple scale of in vitro neuronal networks in order to derive functional measurements indicative of sequential dynamics and synchronization [100]. In general, the analyses of network activity measured with MEA can benefit from these multilevel and modeling approaches.

Finally, network plasticity metrics present an additional avenue for phenotypic characterization by leveraging the ability to stimulate networks through electrical or optogenetic means. Application of focal stimulation to in vitro networks has been shown to induce plasticity-associated alterations in network function and plasticity in primary cultures of rodent neurons, but to our knowledge have not yet been applied in an iPSC disease modeling context [101,102,103,104,105,106,107].

### 5.2. Spike Sorting for Improved Firing Rate Statistics

As analysis techniques improve our ability to draw biologically meaningful interpretations from complex network activity patterns, researchers are afforded an opportunity to improve the depth of network phenotyping by better leveraging the unique strengths of MEA recording technology. Keeping in mind the historical context in which MEAs were developed, these strengths are: (1) the ability to interrogate the electrophysiological relationships between different groups of neurons in culture by recording spatial as well as temporal information about firing activity, and (2) the ability to non-destructively record the same population of neurons over long time intervals to capture how those relationships might evolve as a dissociated neuronal culture develops and self-organizes into an integrated network over multiple weeks in vitro.

A brief survey of the current literature suggests that these strengths have been under-utilized in the iPSC disease modeling studies of neurodevelopmental disorders. The most commonly reported metric for network phenotyping, mean firing rate (MFR), is agnostic to the correlation structure present in network activity datasets and has the potential to characterize many radically different patterns of population-level firing activity as being equivalent (Figure 1). We suggest that MFR-based metrics should be interpreted with caution, and network phenotyping assays should ideally utilize and integrate multiple different activity metrics to capture differences in network behaviour.

Interpreting MFR statistics also becomes a complicated matter when considering that the spike detection algorithms utilized in many commercially available MEA systems rely on threshold crossing methods, which do not discriminate between single-unit and multiunit activity. Consequently, spike trains recorded at individual electrodes may represent the superimposition of spiking activity generated from multiple different neurons firing within the vicinity of the recoding channel (Figure 2A). This can have important implications for spike-level phenotyping metrics such as MFR, or those which describe interspike interval (ISI) distributions. As an illustrative example, in Figure 2B, we have generated single-unit spike trains from 3 imagined neurons which display perfectly periodic firing, with an MFR of 0.5 Hz, mean ISI of 2.0 s, and an ISI coefficient of variation (ISI COV) of zero. However, due to phase differences in the periodic firing of individual units, the combined multiunit spike train presents spike-level statistics that are considerably different from its composite parts.

Thus, firing event statistics in multiunit recordings can be influenced not only by important differences in intrinsic neuronal properties such as membrane excitability and synaptic connectivity, but also by the number of single units contributing to each multiunit signal. Previous work has demonstrated that MFR statistics are highly sensitive to differences in neuronal cell density between cultures [40,108,109]. In addition, recent work by Mossink et al. (2021) found that electrodes located near highly dense, localized clusters of neurons would often detect firing rates significantly greater than other areas of the same culture [40]. It is important to note that substantial differences in cell culture density are known to impact key neuronal attributes including dendrite morphology and synapse density, which likely contribute to changes in MFR statistics and overall network development by directly altering the functional characteristics of individual cells [108,109,110]. However, high variability in firing rate statistics is also observed in experiments where considerable care has been taken to keep cell density relatively consistent between replicate cultures [40]. It is possible that these observations are—at least in part—reflecting some of the inherent limitations of multiunit recordings in capturing more fine-grained differences in the firing repertoire of individual neurons.

Spike sorting methods aim to overcome this limitation by deconvolving multiunit spike signals into their single-unit components. These methods rely on the observation that shapes of extracellular action potential waveforms recorded at a given electrode are often unique for each neuron active at that recording site. Accordingly, single-unit spikes can be “sorted” out from multiunit data by examining the waveform of each spike and grouping those with similar shapes into a putative single unit (Figure 2A) [111]. While spike sorting methods are commonly used during in vivo electrophysiology experiments, they have yet to see widespread adoption by researchers within the in vitro MEA community, potentially owing to the high degree of manual curation and supervision required by many traditional spike sorting methods [112]. In contrast, more recently developed automated spike sorting algorithms (discussed more comprehensively in [113,114,115]) may be better suited for processing the large amounts of data generated by the multi-well MEA formats favoured by many in the iPSC disease modeling field, which contain hundreds or even thousands of recording channels.

Differentiating between single-neuron and multiunit activity may be particularly important for interpreting changes in functional measurements in the context of drug screens and phenotypic rescue assays, such as increases or decreases in the number of spikes detected in a given channel. For example, drugs may act on single-neuron firing via changes in membrane excitability, or at the level of synapses by increasing transmission. In the former scenario, we may observe increases in global spiking activity due to increases in the spiking activity of single neurons. In the latter, we may observe increases in global spiking activity due to a greater number of neurons becoming responsive in a network (Figure 2C). Without using spike sorting methods to differentiate between single neurons and multiunit activity, one cannot determine which scenario (and which underlying biological mechanism) is likely responsible for an observed change in spiking behaviour.

Spike sorting methods may also be useful to reveal interesting aspects of correlated firing between single neurons within a network. Such single-unit analyses may be utilized to capture differences in activation latencies in the scale of milliseconds and reveal directionality of signal flow within a network, as well as measures of spike propagation speed. While such analyses can also be performed using unsorted multiunit data, single-unit analysis may be useful for further probing the interactions and dynamics that arise between different neuronal subtypes in cultures with mixed cell type composition (cultures prepared with different ratios of induced excitatory and inhibitory neurons, for example). Both spike propagation speed and single-unit directional correlation metrics provide more fine-grained insights into intra-network dynamics which could serve as informative phenotypic measures for a variety of neurological disease models.

It should be noted that both manual and automated spike sorting methods are susceptible to sorting errors, such as assigning spikes from different neurons to a single unit (type I errors) or missing the assignment of true spikes to a real unit (type II errors) [116]. These sorting errors can, in turn, impact the findings from downstream analyses such as evaluations of neural correlations, synchrony, and average firing rates [117,118,119]. For example, excessively strict spike sorting criteria will increase the number of type II sorting errors (missed spikes) and result in lower MFRs. While the continual refinement and development of new automated spike sorting methods with reduced error rates may help address these concerns, investigators should nonetheless be mindful of the potential impact that sorting errors may have on their data, and take care to ensure that spike sorting parameters are chosen in a deliberate and purposeful manner. Newer software tools such as SpikeForest and SpikeInterface may be useful starting points for investigators looking to explore spike sorting techniques, as they have streamlined the process of comparing and benchmarking different automated spike sorting algorithms significantly [113,114]. Still, more widespread adoption of spike sorting practices by the iPSC disease modeling community has likely been slowed by a lack of real “plug and play” solutions for seamlessly interfacing automated spike sorting programs with the commercially distributed software suites and proprietary data formats used by many researchers in the field. The development of additional user-friendly tools to address this gap would be of considerable benefit to the community.

## 6. Standardizing MEA Data Reporting

With the inclusion of MEA phenotyping assays becoming more commonplace in iPSC disease modeling studies, there is a need for discussion within the field about best practices when it comes to the reporting and publication of MEA phenotyping data. In our brief survey of the literature, we found considerable variation in how different studies define biological versus technical replicates, as well as the total number of replicates used (Table 1). While most studies surveyed were consistent in defining replicates in terms of the number of independent wells/samples analyzed, we found inconsistent reporting on whether these replicates represent multiple independent wells recorded from a single multi-well MEA plate, or are pooled from multiple plates and/or multiple neuronal differentiations. This is an important distinction, as Mossink et al. (2021) have found significant batch effects between different preparations of neurons and astrocytes, and stressed the importance of including data from multiple neuronal differentiations in phenotyping experiments to account for this technical variability [40]. More specifically, authors were urged to include data from at least two independent neuronal differentiations in their analysis. We would like to echo this statement, and suggest that authors clearly state the number of independent neuronal differentiations and MEA plates used in their analyses when reporting MEA phenotyping results.

Another important consideration for researchers involves the proper statistical handling of MEA phenotyping data. In particular, “shotgun style” phenotyping approaches—which identify phenotypic differences in network behaviour by leveraging commercial analysis software to extract and compare large numbers of network metrics in parallel—should make use of a multiple testing correction when interpreting results. Researchers should be sure to report whether or not this approach was used in their methodology. Our brief survey of the current literature also uncovered discrepancies in whether parametric or non-parametric statistical tests were used to analyze activity metrics such as MFR. Another recent study by Negri et al. (2020) found similar discrepancies in statistical methodology among studies reporting MEA data, and showed that the distribution of MFRs recorded from primary cultures of rat cortical neurons is decidedly non-normal [115]. While this finding may or may not hold true for different cultures of iPSC-derived neurons, it underscores the need for researchers to be vigilant in checking assumptions of normality before proceeding with statistical analysis.

Finally, MEA data reporting could benefit from standardization in how MEA phenotyping data are presented in publication figures. Considering more commonly used experimental techniques such as Western blotting or immunocytological staining for a moment, it has long been standard practice to publish representative images alongside any plots of quantitative metrics such as signal intensities. These representative images both provide some more intuitive understanding of what changes in abstract metrics such as average pixel intensity actually look like, while also allowing other researchers to quickly assess the quality of the raw data presented, ensuring there are no technical artifacts such as high background noise and non-specific western bands which may have impacted the downstream quantification. We believe that a similar standard practice should be adopted for MEA data reporting and suggest that raster plots of representative network activity should always be included as a supplement to any quantitative metrics used to characterize phenotypic differences in network behaviour. Moreover, representative raster plots should display activity from all available recording channels so potential data quality concerns such as large numbers of inactive electrodes or excessively noisy channels (e.g., Figure 1A) can be easily identified. Presentation of representative raw voltage traces and spike waveforms will also help in evaluating data quality, particularly the signal to noise ratio (SNR) of recording electrodes.

The inclusion of representative raster plots is particularly important in studies which include phenotypic rescue assays. The effectiveness of certain drug interventions can easily be misrepresented by selective reporting of only those network metrics which see substantial improvement with drug treatment, even if that improvement does little to change the major qualitative differences in the network activity patterns observed between control and disease states. To illustrate this point, we produced example raster plots by randomly generating spike trains with different firing characteristics, representing 60 s of network activity from an imagined control and disease condition, as well as two hypothetical drug interventions (Figure 3). We generated 20 of these raster plots for each condition, then calculated and plotted the mean firing rate of each (Figure 3F).

Examining this plot in isolation, it can be seen that the mutant condition (MUT) has a significantly reduced MFR when compared to the wild-type (WT) condition. Additionally, it can be seen that Drug A appears to be an effective therapeutic intervention, clearly rescuing the MFR network phenotype, while Drug B and Drug C appear to have no effect. However, a much different story emerges when we include the representative raster plots (Figure 3A–E). We can immediately see that in addition to changes in MFR, the mutant network phenotype is also characterized by a much more dramatic lack of burst firing and an apparent inability to synchronize firing events across multiple recording channels. Moreover, while Drug A may be successful in rescuing differences in MFR, it clearly has no significant impact on the aberrant firing organization that is also characteristic of the mutant condition. As such, it would be difficult to argue that this compound is effective in coercing the mutant networks to fire in a more “wild type-like” manner. In contrast, we see that Drug C does appear to be successful in this task, despite being written-off as having no effect when MFR is taken to be the only phenotypic metric of importance.

While the inclusion of representative raster plots in MEA phenotyping studies is clearly a step in the right direction towards ensuring high standards of scientific integrity and transparency in data reporting, we also believe that a greater effort should be made to make all raw MEA recording files freely available for other researchers to utilize. This has been a difficult task for the field, as current multi-well MEA systems are capable of generating multiple gigabytes of data per minute of recording. Large phenotyping studies can therefore generate multiple terabytes worth of raw data, and there is currently a lack of a convenient platform for storing and curating these large recording files. We would like to suggest that the creation of a public MEA data repository should be a goal of the field.

Finally, an issue with the analyses of MEA data is that commonly used electrophysiological analyses techniques such as spike sorting, multidimensional analyses of data, and machine learning are not part of the training repertoire of developmental and molecular biologists. A possible solution to this problem is to train the new generations of researchers in a diversity of data analyses and visualization techniques for electrophysiological data. Another possible solution is to promote more collaboration between laboratories with different levels of expertise. Our proposal of generating platforms for making MEA data available to the community could aid in that respect.

## 7. Conclusions and Recommendations

MEA technology stands to yield new insight into disease phenotypes by revealing population-level differences in neuronal firing and communication. As others have noted, MEA phenotyping assays can be highly robust so long as careful consideration is given to proper assay design, analysis, and data reporting. In this review, we have highlighted some of the current challenges with standardizing MEA experiments within the iPSC disease modeling field and provided some recommendations for working towards this goal.

### 7.1. Considerations for Experimental Design

In examining assay design options across recent studies in the field, we first noticed that the most common choices of recording durations were 5 and 10 min (Table 1). This is considerably shorter than recording durations commonly used in primary rodent neuron literature which often range between 20 and 60 min, with some exceptions [115]. Given previous observations that cultures may transition between different “modes” of network firing activity (e.g., M5–M4–M5) during long-term recordings, increasing the duration of MEA recording sessions could be of benefit to the iPSC disease modeling community. For example, Wagenaar et al. (2006) reported that many MEA cultures appeared to go through a developmental period where the temporal organization of firing activity was characterized by the appearance of so called “superbursts”, defined as clusters of rapid network bursting activity separated by several minutes of tonic spiking with no network bursts [109]. In such firing regimes, a short recording duration of 2–5 min may be inadequate to reliably capture the complete pattern of network activity being generated. Thus, the question of what should be considered an appropriate recording duration may require some more careful consideration from the field. Negri et al. (2020) addressed this question for cultures of primary mouse cortical neurons by recording mature cultures on two consecutive days, then calculating Pearson’s correlation between the MFRs observed on day one versus day two for increasingly longer time bins [115]. The point at which the correlation of MFRs no longer increased significantly with increased recording duration (in this case, 30 min) was chosen as an appropriate recording interval. We suggest that a similar procedure should be used to determine suitable recording times for cultures of iPSC-derived neurons.

We also noticed some discrepancies in how biological replicates were defined and reported in the literature. While we found that most studies defined replicates in terms of the total number of culture wells analyzed, it is important to note that considerable variation in the development of networks from different culture platings and different neuronal differentiations has been reported in both iPSC and primary neuron literature [18,40,109]. It is therefore imperative that researchers utilize multiple independent neuronal differentiations, plated on independent MEA plates, in their assay design. Moreover, the total number of neuronal preparations and MEA plates used should be clearly stated in any research publication. We support recommendations made by Mossink et al. (2021) that investigators should use at least two batches of neurons, preferably paired with the same batch of supporting astrocytes [40].

Final considerations for assay design concern important parameters which must be controlled to avoid introducing unwanted variability and noise into recordings. Spontaneous network activity has been shown to be sensitive to changes in environmental factors including media pH and osmolarity, ambient temperature, as well as physical movement or vibration of the MEA plate or recording apparatus [18,109]. Many modern MEA systems have recording chambers that carefully regulate environmental conditions including temperature, CO_2_ saturation, and relative humidity to help mitigate these concerns; however, care should be taken to schedule recordings around culture media changes to avoid introducing variability from pH differences between fresh and old culture media. Cultures should also be given 5–10 min to baseline activity levels after being physically moved from an incubator to the recording station. In addition, if a transcription factor programming-based differentiation method is being used to generate iPSC-derived neurons, care should be taken to ensure that similar levels of transgene expression are being induced in control and affected cell lines, as differing levels of *NGN2* expression have been shown to promote the adoption of different neuronal cell fates [120]. As best practice, any iPSC lines being compared using lentivirus transgene delivery should be transduced with the same titre of virus, and be at similar post-infection passage number at day 0 of differentiation. These measures help control for large differences in transgene copy number and transgene silencing effects, respectfully.

### 7.2. Considerations for MEA Data Analysis

In this review, we advocate for a more holistic approach to network phenotyping—one which places greater emphasis on describing differences in the spatiotemporal organization of spontaneous firing into characteristic activity patterns rather than focusing on aggregate firing rate differences alone. Using didactic examples, we illustrate how simple activity metrics such as MFR can fail to capture significant changes in the arrangement of firing activity within a recording (Figure 1), and how this can potentially cause issues in the interpretation of phenotypic rescue assays (Figure 3). While it is difficult to recommend a set of phenotypic metrics which will always be informative in all cases, our most basic recommendation is to ensure that multiple assay readouts are used to adequately describe network activity. Ideally, these should include metrics which capture some aspect of temporal organization, such as bust, network burst, and ISI descriptors. Interestingly, we noted that very few studies in our survey of current literature reported synchrony metrics as part of their assay design (Table 1). As synchrony metrics reflect a degree of both spatial and temporal structure in network-wide firing patterns, including a synchrony analysis alongside basic activity metrics such as MFR may be a relatively simple way to capture more detailed information about network dynamics. More advanced computational methods such as network modeling, state space, and functional topology analyses provide further avenues for deeper characterization and investigation of network properties, and provide a good incentive for investigators to seek out new collaborations with the fields of computational and mathematical neuroscience.

### 7.3. Considerations for Data Reporting

Our final considerations concern how MEA data are reported and presented in publications. While many commercial MEA analysis software suites are capable of automatically extracting dozens of different activity metrics in parallel, we noticed that few studies disclosed whether this approach to network phenotyping was used or not, nor whether any additional metrics were investigated but omitted from the final publication. We believe this has important implications for downstream statistical analysis, specifically whether one needs to correct for multiple comparisons, and authors should explicitly state whether this approach was used when describing their methodology.

As a standard, we suggest that representative raster plots, raw signal traces, and spike waveforms should always be included alongside any summary plots of MEA phenotyping metrics. We believe this will allow reviewers and other investigators to easily evaluate the quality of the raw recording data, and is particularly important for evaluating claims about the efficacy of different drug compounds in phenotypic rescue experiments (Figure 3). Finally, we would like to end by reiterating the need for a public repository for raw MEA recordings to allow for both more transparent reporting and easier sharing of large data files between research groups.

## Figures and Tables

**Figure 1 biology-11-00316-f001:**
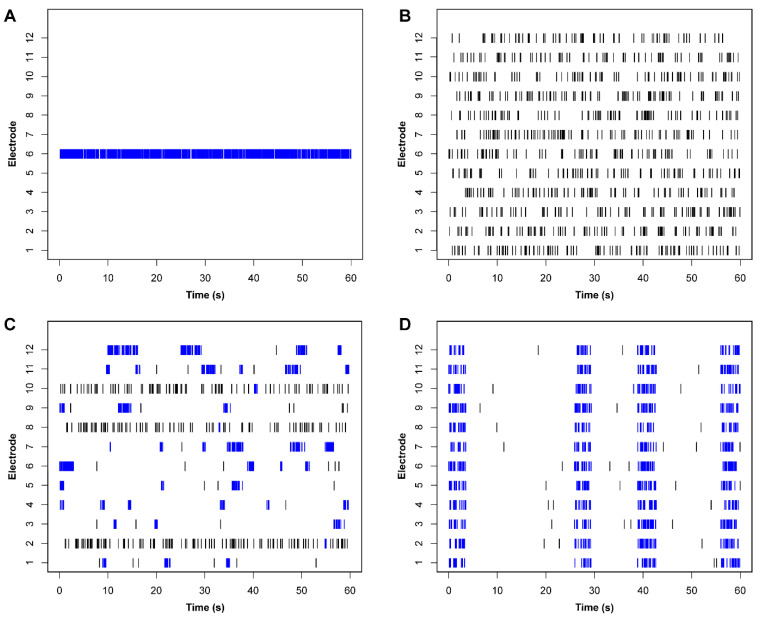
Different spatiotemporal organization of spiking activity in networks with identical mean firing rates. In this didactic example, each figure panel shows a representative raster plot from four hypothetical networks with identical mean firing rates of 20 Hz (1200 spikes in 60 s). (**A**) A network with very little spatial distribution of activity; all 1200 spikes are recorded from a single electrode in the array. (**B**) Spiking activity is well distributed across all 12 electrodes, with activity primarily occurring as loosely organized tonic spiking (black) rather than clustered burst firing (blue). (**C**) A network with a mixture of tonic and burst firing. Low synchronization of bursts across multiple electrodes. (**D**) Spiking activity is highly organized into tightly synchronized network bursts.

**Figure 2 biology-11-00316-f002:**
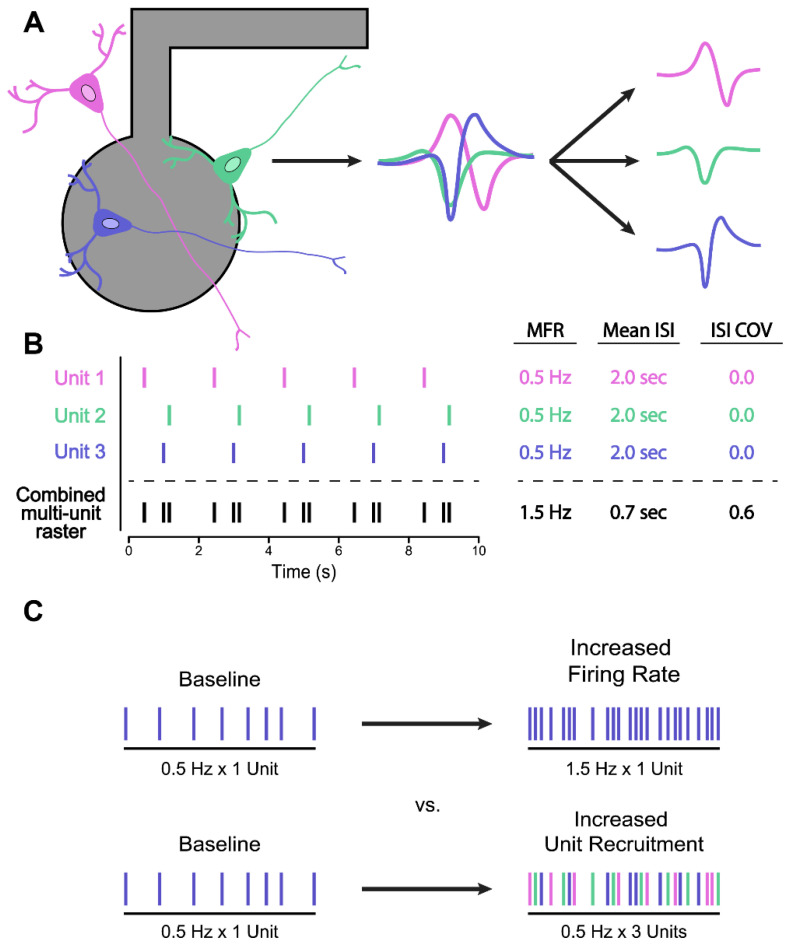
Spike sorting for more accurate spike-based phenotyping metrics. (**A**) Multiple neurons within recording range of a single extracellular electrode generate action potentials with unique waveform shapes. Note that dendrites and axons (thin longer lines) are not drawn to scale and would extend further than illustrated. By grouping spikes with similar waveform shapes into different clusters, spike sorting algorithms can separate multiunit spike trains into putative single-unit spike trains. (**B**) In this example, an unsorted multiunit spike with a MFR of 1.5 Hz and a mean ISI of 0.7 s is composed of three single units, each with a lower MFR of 0.5 Hz and a greater mean ISI of 2 s. (**C**) An example of how spike sorting may help differentiation between true MFR changes (**top**) versus changes in the number of active units that have been recruited into a network (**bottom**).

**Figure 3 biology-11-00316-f003:**
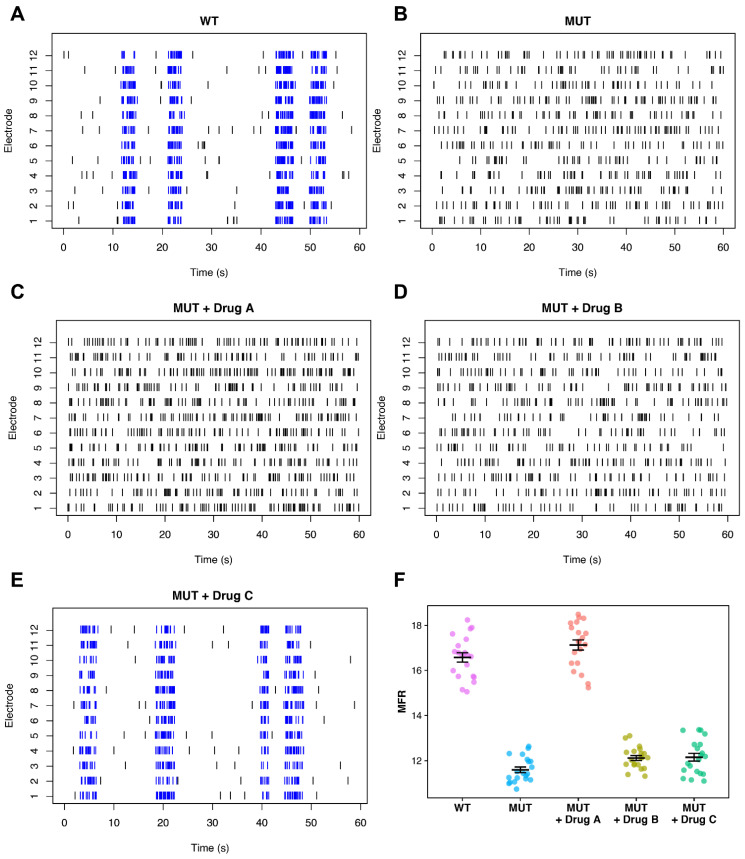
Importance of representative raster plots in phenotypic rescue experiments. Representative raster plots from hypothetical wild-type (**A**) and mutant (**B**) networks, as well as mutant networks treated with three theoretical drug compounds (**C**–**E**). (**F**) Quantification of mean firing rates for each group.

**Table 1 biology-11-00316-t001:** Overview of MEA assays in recent iPSC disease modeling studies. ASD = autism spectrum disorder, TSC = tuberous sclerosis complex, FXS = fragile X syndrome, SCZ = schizophrenia, ADHD = attention deficit hyperactivity disorder, Rett = Rett syndrome, MFR = mean firing rate, wMFR = weighted mean firing rate, ISI COV = interspike interval coefficient of variation, IBI = interburst interval, and IBI COV = interburst interval coefficient of variation.

Reference	Differentiation Type	Disease Model	Associated Mutations	System and Plate Format Used	Number of Electrodes	Reported Metrics	Recording Duration	Replicates per Line
Russo et al. (2018) [22]	Directed	ASD	SETD5, Idiopathic	Axion Biosystems 12-well	64	MFR	3 min	6
Deneault et al. (2019) [23]	TF Programming (*NGN2*)	ASD	CTN5, EHMT2, DLGAP2, CAPRIN1, SET, GLI3, VIP, ANOS1, THRA, NRXN1, AGBL4	Axion Biosystems 48-well	16	wMFR, network burst frequency	5 min	9–24
Deneault et al. (2018) [24]	TF Programming (*NGN2*)	ASD	ATRX, AFF2, KCNQ2N SCN2AM ASTN2	Axion Biosystems 48-well	16	MFR, burst frequency, network burst frequency	5 min	21–55
Marchetto et al. (2017) [25]	Directed	ASD	-	Axion Biosystems 12-well	64	Number of spikes, network burst frequency	10 min	3
DeRosa et al. (2018) [26]	Directed	ASD	-	Axion Biosystems 12-well	64	MFR	10 min	16
Amatya et al. (2019) [27]	Directed	ASD	-	Axion Biosystems 96-well	8	Minimum embedding dimension, ISI COV	10 min	6
Winden et al. (2019) [28]	TF Programming (*NGN2*)	TSC	TSC2	Axion Biosystems 48-well	16	wMFR, synchrony index	-	48
Nadadhur et al. (2019) [29]	Directed	TSC	TSC1, TSC2	Multi Channel Systems single well	60	MFR	10 min	6–8
Quraishi et al. (2019) [30]	Cellular Dynamics (proprietary)	Epilepsy	KCNT1	Axion Biosystems 48-well	16	MFR, Synchrony Index, burst rate, burst duration, burst intensity	8 min	24
Graef et al. (2020) [31]	TF Programming (*NGN2*)	FXS	FMR1	Axion Biosystems 48-well	16	wMFR	5 min	12–24
Liu et al. (2018) [32]	Directed	FXS	FMR1	Axion Biosystems 12-well	64	MFR	5 min	2–6
Utami et al. (2020) [33]	Directed	FXS	FMR1	Axion Biosystems 12-well	64	MFR, max firing rate, number of unresponsive	5 min	6
Nageshappa et al. (2016) [34]	Directed	MECP2 duplication syndrome	MECP2	MED64 single well	64	Network burst frequency	5 min	3
Kathuria et al. (2019) [35]	Directed	SCZ	-	MED64 12-well	16	MFR	1 min	3
Sarkar et al. (2018) [36]	Directed	SCZ	-	Axion Biosystems 96-well	8	Number of spikes, Synchrony Index, Burst Frequency, Network Burst Frequency	10 min	6 or 12
Ishii et al. (2019) [37]	TF Programming (NGN2 or *ASCL1* + *DLX2*)	SCZ and Bipolar	idiopathic, PDH15, RELN	Axion Biosystems 48-well	16	wMFR, GABA Sensitivity	5 min	4–6
Sharma et al. (2019) [38]	Directed	Rett	MECP2	Axion Biosystems 12-well	64	Network burst frequency	5 min	3
Frega et al. (2019) [39]	TF Programming (*NGN2*)	Kleefstra syndrome	EHMT1	Multi Channel Systems 24-well	12	MFR, burst frequency, burst duration, mean IBI, IBI CV, % spikes out of bursts	20 min	10–23
Mossink et al. (2021) [40]	TF Programming (NGN2 or ASCL1 + forskolin)	ASD, ADHD	CHD13	Multi Chanel Systems 24-well	12	Network burst duration, number of spikes per network burst	10 min	20–49
Alsaqati et al. (2020) [41]	Directed	TSC	TSC2	Axion Biosystems 24-well	16	MFR, network burst frequency, network burst duration, inter-network burst interval, burst frequency, connectivity correlation, % spikes outside network bursts, frequency distribution	-	3–10
Li et al. (2013) [42]	Directed	Rett	MECP2	MED64 single well	64	MFR	5 min	-
Sundberg et al. (2021) [43]	Directed	16p11.2 CNV	16p11.2 dup, 16p11.2 deletion	MaxWell Biosystems single well Axion Biosystems 48-well	26,400 16	MFR, fraction of synchronized sensors, burst frequency, burst duration, inter-burst interval, number of spikes per burst	2 min 5 min	4–7 6–16

## Data Availability

Not applicable. No new data were created.

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
