# Peer review of "Multielectrode Arrays for Functional Phenotyping of Neurons from Induced Pluripotent Stem Cell Models of Neurodevelopmental Disorders"

_biology, 2022, doi:10.3390/biology11020316_

Round 1

Reviewer 1 Report

The authors described the history of MEA, and the current associated studies and proposed the data repository for the future studies. This review manuscript is very interesting and challenging, but very precisely described. Especially, the figures help the readers understand the content. 

I have one comment. In page 13, the authors should explain what is MED.

Author Response

REVIEWER 1

The authors described the history of MEA, and the current associated studies and proposed the data repository for the future studies. This review manuscript is very interesting and challenging, but very precisely described. Especially, the figures help the readers understand the content. I have one comment. In page 13, the authors should explain what is MED.     

We thank the reviewer for their comments! We have more clearly defined MED (minimum embedding dimension) in this section.

Reviewer 2 Report

In the presented review Multielectrode Arrays for Functional Phenotyping of Neurons from Induced Pluripotent Stem Cell Models of Neurodevelopmental Disorders, authors point to a variety of challenges with handling and analysis of MEA recordings of in vitro IPSC cultures. The authors urge to prioritize activity patterns over simple firing metrics and emphasize the necessity for a holistic and consistent analysis in IPSC-based research. They have rightfully pointed to inconsistencies in previous research, from measurement approach to data analysis and presentation. The manuscript writing is clear. 

In particular the authors point out the value of MEA measurements being introduced to the iPSC community. The authors are particularly correct that a standard in the field for parameters to be reported, data availability, and presentation would optimize efficiency and avoid the difficulties comparing across publications that has been experienced by the primary neuron community using MEAs. To that end, a more in depth discussion of the standards in primary neurons may accelerate the iPSC community to standards more effectively. What lessons are to be learned from how primary neuron literature has evolved. One may also comment on the lack of coherence in characterization and definition of parameters in primary neuronal networks. In such a light, the remaining need to standardize in the iPSC derived neurons on MEA field may be seen as a historical bad practice from the neuron community. There is also room for comment on how, in the face of these inconsistencies, one can best compare iPSC derived neuron publications to primary neuron publications. See for example commentary in PMID 27098024.

Furthermore, a number of concepts for this community standard are mentioned (see also comments to line 377 and 600), but the authors do not summarize their final assertion of their suggested standards considering the reviewed material. For example, reverting events are mentioned - how long should a community standard measurement be to ensure events like that are not missed? If a researcher desires to adopt this technique for iPSCs after reading this review, what parameters are critical for their decision which equipment to invest in? Furthermore, which critical parameters must be controlled (spatial resolution of recording, temperature, pH, SNR, ...)?

The review offers a variety of analysis approaches, however, the advancement of a holistic, systemic approach in network analysis is missing. Authors overlook major approaches in computational neuroscience which qualitatively describe functional organization within the network. While the reported CVnetwork and CVtime parameters partially describe the wholistic network behavior, the downstream steps in network analysis, which serve to estimate functional network topology, and its changes over time, are missing. Authors should refer to the review of Poli et al., 2015, and the work of Downes et al., 2012. Similar to the network models that are reported in this review, estimation of global network parameters reduces dimensionality, but without the simulation steps: these parameters are inferred from the activity itself. Authors should generally separate firing metrics from synchronicity metrics and global network parameters (if implemented). 

In the spike sorting section, authors should emphasize the type I and type II sorting errors, present both in automatic and manual spike sorting. These errors arise due to the intrinsic intracellular action potential waveform variability during different modes of activity, and can seriously affect the estimations of network parameters. Authors should refer to the work of Henze and Buzsaki.

Comments to specific passages by line number:

102: ‘reaggregation’ to in vitro neuroscientists might imply the formation of neuronal clusters (aggregates), which is usually an unwanted effect due to poor adhesion. Consider rephrasing it into something like ‘re-establishment of de novo connections’.

198-199:  These reverting events may add noise in qualitative estimations, or they may uncover a higher level of network dynamic changes. If some cultures tend to go M5-M4-M5, whereas others stay on M5 after X days in culture, these are valuable findings and should be also emphasized.

243: Please also define the disease model abbreviations for Table 1.

377: This sounds like the introduction to what those MEA parameters that ought to be standardized across the field should be. If not listed here, can the further discussion be cross-referenced?

469: Undefined abbreviation MED – presumably minimum embedding dimension.

513: 'type' seems it should be 'time'.

512-518: Noting that these methods are used in vivo and in organoids, it would be useful to comment on their current state in vitro or explain the current lack of their use in vitro relative to the proposed transfer to iPSCs in vitro.

594-596: Latencies and weighted directionality based on cross-correlation of spike timings can be inferred between multiple electrodes without the spike sorting. Furthermore, how does directionality relate to the proposed use? If drug screening or comparing one random in vitro culture to a control random in vitro culture – is the directionality or the information propagation speed the more relevant parameter?

600-612: This is a critical point that warrants more discussion. As the authors will hopefully present their suggestion for parameters to fingerprint in the culture characterization, a lower limit on replicates and commentary on the number of tests necessary to be relevant should be included.

Figure 2, there is well established literature suggesting which wave forms can predominantly be expected from which neuronal compartments. In all 3 of the neurons sketched, the electrode is recording from axon, AIS, and the axon end of the soma and these would be the most likely to be falsely attributed as the same unit. Furthermore, the relative size of the electrode to the proposed neurons is too big for attributing single units.

637-651: This reviewer fully supports the authors’ suggestion to always publish representative MEA data to support statistical metrics. Even to one step further, raw traces, with a presentation that shows both spike activity and waveform shape should be a standard inclusion for evaluating the quality of data that statistical parameters were generated from.
